# Comparative Transcriptome Analysis Explores the Mechanism of Angiosperm and Gymnosperm Deadwood Degradation by *Fomes fomentarius*

**DOI:** 10.3390/jof10030196

**Published:** 2024-03-04

**Authors:** Yulian Wei, Jianbin Xue, Jiangtao Shi, Tong Li, Haisheng Yuan

**Affiliations:** 1Key Laboratory of Forest Ecology and Management, Institute of Applied Ecology, Chinese Academy of Sciences, Shenyang 110016, China; xuejianbin22@mails.ucas.ac.cn; 2University of Chinese Academy of Sciences, Beijing 100049, China; 3College of Materials Science and Engineering, Nanjing Forest University, Nanjing 210037, China; shijt@njfu.edu.cn; 4School of Environment and Science, Centre for Planetary Health and Food Security, Griffith University, Nathan, Brisbane, QLD 4111, Australia; tong.li7@griffithuni.edu.au

**Keywords:** *Fomes fomentarius*, deadwood degradation, comparative transcriptome analysis, host preference mechanism

## Abstract

In forest ecosystems, most of the soil organic matter is derived from trees, as deadwood lignocellulose and wood-decaying basidiomycetes are the most important decomposers of lignin and cellulose. *Fomes fomentarius* is one of the most common white-rot fungi colonizing angiosperm trees and is often found in birch deadwood but seldom in pine deadwood. To reveal the mechanism through which *F. fomentarius* selects angiosperms as its preferred host trees, birch and pinewood sticks were selected for culturing for two months. The weight loss, cellulose and lignin degradation rates, activities of degrading enzymes, and transcriptome analyses of two degradation models were compared and analyzed. The results showed that *F. fomentarius*-degraded birchwood with higher efficiency than pinewood. A GO enrichment analysis found that more upregulated genes related to the top 30 terms showed a molecular function related to degradation, and most genes belonged to the CAZymes family in *F. fomentarius*-degraded birchwood. However, pinewood degradation did not show these phenomena. A KEGG pathway analysis also indicated that, for the same pathway, more upregulated genes were involved in birchwood degradation caused by *F. fomentarius* than in pinewood degradation.

## 1. Introduction

Trees are one of the most precious resources on Earth and important carbon pools, which including their growth, death, and composition. The cell walls in wood consist of three main polymers: cellulose, hemicelluloses, and lignin. Among these, cellulose is an important natural component, with the most abundant of it occupying about 40% of the dry weight of woody biomass [1]. Wood-decaying fungi are the most important fungal group and can decompose lignin and cellulose in deadwood by producing various degrading enzymes. Of all wood-rotting basidiomycete fungi, more than 90% can cause white rot [2]. They are widely distributed in nature in huge abundance and are equipped with the ability to completely mineralize all lignocellulosic cell-wall components and convert them into carbon dioxide and water, which makes them key players in the recycling of carbon in nature. They have attracted considerable attention, and notable research work includes a thorough study of wood decomposition [3]. It has been reported that white-rot fungi have developed a capacity to penetrate the cell wall between wood elements directly within wood tissues. Fungi accomplish this by producing slender hyphae that can bore through the cell wall through localized cell-wall degradation. Hydrolase and oxidase play a key function during penetration. The wood-decaying fungal enzymology of lignocellulose-decomposing extracellular enzymes results in the white-rot mode, where lignin and polysaccharides are both utilized [4,5,6,7].

*F. fomentarius* is a common wood-rotting fungus in deciduous forests in temperate areas [8] and causes white rot. In China, it is often found in birch (*Betula platyphylla*) and hardly in gymnosperm wood [9]. This fungus has a wide niche; it colonizes a living tree as a parasite, reaches deadwood, and, in this way, decomposes it using its significant combative abilities [10,11]. It has the important ecological function of pre-colonizing wood, thus having a stimulatory effect on other decaying fungi, which are secondary colonizers [12]. In nature, it is often found that this fungus can dominantly colonize large volumes of wood (although its cohabitation with other fungal species cannot be excluded) and plays a key role in the carbon cycle. However, the degradation mechanism of *F. fomentarius* in wood is not clear. As an important white-rot fungus, it needs to be revealed urgently. 

In recent years, transcriptome profiling using RNA-seq has rapidly developed, and there is increased research relevant to fungi [13,14]. Here, a transcriptome analysis of *F. fomentarius*-degraded angiosperm and gymnosperm deadwood was carried out. *Betula platyphylla* Sukaczev and *Pinus koraiensis* Siebold and Zucc. are the dominant tree species of the natural temperate forest in the Changbaishan Nature Reserve, Northeastern China. In nature, *F. fomentarius* can be found in birch, from living trees to fallen wood, but it has not been found or reported in pine. These two tree species were studied as the representatives of angiosperms and gymnosperms, separately. By comparing the discrepancy between *F. fomentarius*-degraded birchwood (abbreviated as FFB) and pinewood (abbreviated as FFP), we attempted to uncover the answers to the following questions: Can *F. fomentarius* degrade gymnosperm trees? Why does it prefer to choose hardwood trees over coniferous trees under natural conditions? Is there a difference in the mechanism between these two tree types? These answers will help us understand the selection mechanism of *F. fomentarius* in nature and its ecological function.

## 2. Materials and Methods

### 2.1. F. fomentarius Cultivation and Woody Stripe Degradation

A pure strain of *F. fomentarius* (No. Wei 8498) was isolated from fruitbodies that were collected from fallen birch deadwood in Changbai Mountain Nature Reserve, China, on 17 September 2018. The hyphal culture medium contained 2% malt extract powder, 0.8% agar powder, and 1 L of water. Hexagonal bottles with 800 mL capacities were used in these culturing experiments, covered with sealing film to maintain the atmosphere. Pine sapwood and birch sapwood from trunks were cut into strips of ca. 6 cm in length, 1 cm in width, and 1 cm in height (Figure 1A➀,➁). These wood strips were weighed, numbered, and recorded after drying at 60 °C. Then, they were soaked and sterilized at 121 °C for 1 h. To obtain a uniform spread of the hyphae in the medium, the packages were kept at 26 °C in the dark. After the mycelium had colonized the medium completely within 7 days, the sterilized pine and birchwood strips were put into the culturing bottles separately; 6 numbered strips per bottle were arranged to weigh over 60 days. Culturing bottles with only hyphae without wood strips was the control, abbreviated as FFC. Hyphae RNA samples were numbered CW8498-1, CW8498-2, and CW8498-3. Culturing bottles with birchwood strips was abbreviated as FFB. Hyphae RNA samples were numbered WW8498B1, WW8498B2, and WW8498B3. Culturing bottles with pinewood strips was abbreviated as FFP. Hyphae RNA samples were numbered WW8498P1, WW8498P2, and WW8498P3. Three replicates were taken for each treatment. 

### 2.2. Weight, Cellulose, and Lignin Loss of Wood Stripe Degradation 

After culturing at 26 °C in the dark for 60 days, the wood strips, hyphae from wood strips, and hyphae from pure medium were taken out of the culturing bottles separately. Six wood strips from three bottles (two wood strips per bottle) were weighed after cleaning hyphae, and the weight loss was calculated. One wood strip of pine and birch was scanned using an electron microscope. Other strips were shattered into sawdust using a Bear FSJ-A03D1 (Bear Electric Appliance Co., Ltd., Foshan, China) grinder, sieved by a 40-mesh sieve for cellulose and lignin measurement and degradation enzyme activities determination. The cellulose and lignin content were measured by Van Soest [15] and improved with filter bags. Samples from hyphae growing in birchwood are named FFB; samples from hyphae growing in pinewood are named FFP; and hyphae from pure medium are named FFC in the following text. 

### 2.3. RNA Extraction, Library Construction, and RNA-Seq 

Total RNA was isolated using the Trizol Reagent (Invitrogen Life Technologies, Carlsbad, CA, USA). Then, concentration, quality, and integrity were determined using a NanoDrop spectrophotometer (Thermo Scientific, Waltham, MA, USA). Three micrograms of RNA were used as input material for the RNA sample preparations. Sequencing libraries were generated using a TruSeq RNA Sample Preparation Kit (Illumina, San Diego, CA, USA). Briefly, mRNA was purified from total RNA using poly-toligo-attached magnetic beads. Fragmentation was carried out using divalent cations at elevated temperatures in Illumina’s proprietary fragmentation buffer. The first strand of cDNA was synthesized using random oligonucleotides and SuperScript II. Second-strand cDNA synthesis was subsequently performed using DNA Polymerase I and RNase H. The remaining overhangs were converted into blunt ends via exonuclease/polymerase activity, and the enzymes were removed. After adenylation of the 3′ ends of the DNA fragments, Illumina PE adapter oligonucleotides were ligated to prepare for hybridization. To select cDNA fragments of the preferred 200 bp in length, the library fragments were purified using an AMPure XP system (Beckman Coulter, Beverly, CA, USA). DNA fragments with ligated adaptor molecules at both ends were selectively enriched using Illumina PCR Primer Cocktail in a 15-cycle PCR reaction. Products were purified (AMPure XP system) and quantified using the Agilent high-sensitivity DNA assay on a Bioanalyzer 2100 system (Agilent Technologies Inc., Santa Clara, CA, USA). The sequencing library was then sequenced on a Hiseq platform (Illumina) by Shanghai Personal Biotechnology Co., Ltd., Shanghai, China.

### 2.4. Basic Analysis and mRNA Analysis of F. fomentarius Transcriptome

Samples were sequenced on the platform to obtain image files, which were transformed by the sequencing platform software, and the original data in FASTQ format (raw data) were generated. Sequencing data contain several connectors and low-quality reads, so we used Cutadapt (v1.15) software to filter the sequencing data and obtain high-quality sequences (Clean Data) for further analysis. In data mapping analysis, the reference genome and gene annotation files were downloaded from the genome website (https://mycocosm.jgi.doe.gov/Fomfom1/Fomfom1.home.html, accessed on 7 January 2022). The filtered reads were mapped to the reference genome using HISAT2 (http://ccb.jhu.edu/software/hisat2/index.shtml, accessed on 7 January 2022), where the default mismatch was no more than 2. The alignment region distribution of mapped reads was calculated. The HTSeq (0.9.1) statistical framework was used to compare the read count values on each gene as the original expression of the gene, and then FPKM was used to standardize the expression. Then, the genes with differential expression under the screened conditions were analyzed with DESeq (1.30.0) as follows: expression difference multiple |log2FoldChange| > 1, significant *p*-value < 0.05. At the same time, bi-directional clustering analysis of all different genes of samples was performed using the R language Pheatmap (1.0.8) software package. Heatmaps could be obtained according to the expression levels of the same gene in different samples and the expression patterns of different genes in the same sample, using the Euclidean method to calculate the distance and the Complete Linkage method for clustering. Next, all the genes were mapped to terms in the Gene Ontology database, and we calculated the numbers of differentially enriched genes in each term. Based on the whole genome, terms with significant enrichment of differentially expressed genes were calculated using hypergeometric distribution. The purpose of GO enrichment analysis is to obtain GO functional terms with significant enrichment of differentially expressed genes, thus revealing the possible functions of differentially expressed genes in the samples, as well as counting the number of differentially expressed genes at different levels of KEGG pathways; then, it is possible to determine the metabolic pathways and signaling pathways in which the differentially expressed genes mainly participate. 

### 2.5. Degradative Enzyme Activities

Crude enzyme extraction: put 1.5 ± 0.05 g woody sawdust samples into a 150 mL conical flask, then add 15 mL 50 mmol/L phosphate buffer solution and seal. Put them in a shaker for 2 h at 110 rpm, 26.5 °C. Transfer the sample to a centrifuge tube and centrifuge at 4000 rpm for 10 min, then filter supernatant fluid with Whatman # 42 filter paper (General Electric System Co., Ltd., Boston, MA, USA). The filtrate is the crude enzyme solution, which can be stored temporarily in a refrigerator at 4 °C.

Enzyme measuring: the activities of degradative enzymes were determined in 6 samples, including *F. fomentarius*-degraded birch and pine sticks, respectively. In the aqueous extracts of the milled samples, lignin-related oxidoreductases (Laccase, EC1.10.3.2) were assayed spectrophotometrically with ABTS (2,20-azino-bis (3-ethylbenzothiazoline-6-sulfonic acid). The activities of cellulose enzymes, i.e., Endo-1,4-β-glucanase (EC 3.2.1.4) and Endo-1,4-β-xylanasein (EC 3.2.1.8), were measured spectrophotometrically by applying AZO-cellulose and AZO-xylan, respectively. The activities of hydrolytic enzymes (β-1,4-glucosidase, EC 3.2.1.21; β-1,4-xylosidase, EC 3.2.1.37; cellobiohydrolase, EC 3.2.1.91; 1,4-β-*N*-acetylglucosaminidas, EC 3.2.1.52) were determined spectrophotometrically by applying *p*-nitrophenyl-b-D-glucoside, *p*-nitrophenyl-b-D-xyloside and *p*-nitrophenyl-*N*-acetyl-b-D-glucosaminide, respectively [16,17]. Activities were photometrically measured in the aqueous extracts in 96-well plates (F-bottom plates; Greiner Bio-One GmbH, Rickenhausen, Germany) with a plate reader (In-finite M200, Tecan, Männedorf, Switzerland). One unit of enzyme activity was defined as the amount of enzyme forming 1 nmol of reaction product per minute [16]. All measurements were taken in 3 replicates. For further analyses, the mean values were used. 

### 2.6. Statistical Analysis 

Sequences obtained with next-generation sequencing were deposited in the Sequence Read Archive (SRA) database in NCBI (https://submit.ncbi.nlm.nih.gov, accessed on 19 October 2023) (SRA accession number: PRJNA1029195). Statistical analyses were conducted in R 4.2.0 [18]. All raw data were analyzed through repeated tests, and significant differences (*p* < 0.05) and extremely significant differences (*p* < 0.01) were analyzed.

## 3. Results

### 3.1. Wood Stick Degradation

Over 60 days, many hyphae of *F. fomentarius* grew vigorously and covered on the wood strips (Figure 1A➂), and the shape of wood sticks changed obviously (Figure 1A➃). The wood structure was significantly destroyed. The cell wall of wood can be observed collapsed and broken with an electron microscope. The mycelia grew inside the wood in the space (lumen) among deadwood cells (Figure 1A➄–➇). The dry weight of birchwood sticks decreased by 35.42%, and that of pinewood sticks decreased by 13.17% (Figure 1B). The cellulose content in birchwood decreased from 0.42 g to 0.20 g per 1 g wood samples, a 0.22 g decrease, on average, while cellulose content in pinewood decreased from 0.46 g to 0.31 g per 1 g origin wood samples, 0.15 g decrease on average. Lignin content in birchwood decreased significantly, from 0.40 g to 0.13 g per 1 g origin wood samples, a 0.27 g decrease, on average, while in pinewood, this varied from 0.36 g to 0.23 g, a 0.13 g decrease per 1 g origin wood samples, on average (Figure 1B). 

Furthermore, in the experiment where *F. fomentarius* decayed wood sticks for 60 days, seven degradative enzyme activities were found to be different between the two tree species separately. The activity of β-1,4-glucosidase, 1,4-β-*N*-acetylglucosaminidase, cellobiohydrolase, and endo-1,4-β-glucanase in birchwood sticks was higher than in pine. However, laccase activity in pine (1986.6 nmol/g/min) was approximately four times higher than that in birch (445.5 nmol/g/min) (Figure 1C).

### 3.2. Data Quality Control Analysis of Transcriptome Sequencing

To understand the effect of different host wood types on the transcriptional levels of *F. fomentarius*, RNA libraries of *F. fomentarius* were built. After high-throughput sequencing, the libraries generated had 43 111 271 (FFB), 44 140 919 (FFP), 45 591 481 (FFC) clean reads from paired-end reads with a single-read length of 150 bp and Q30 percentages (percentage of sequences with sequencing error rates < 1%) and mean GC content of 47%. These data showed that the throughput sequencing quality was high enough to warrant further analysis.

### 3.3. Analysis of Differential Expression of mRNA 

In this study, the differential expression of genes was analyzed by DESeq2 (repeated samples). There were significant differences in gene expression between FFP and FFB after two months. The results of genes with significant differences in different treatments are shown in Table 1. The upregulated or downregulated genes are defined as those with significantly increased (up) or decreased (down) expression compared with the control. A *p*-value < 0.05 was selected after ANOVA tests according to the screening criteria of differential genes (*p*-value corrected using the Benjamini–Hochberg procedure), and the absolute value of the difference in gene expression was more than 1. In addition, the data of differential genes were obtained as the basis for screening. The results showed that 761 genes were significantly upregulated and 636 genes were significantly downregulated in FFB compared with FFC. On the other hand, in FFP, 523 genes were significantly upregulated, and 437 genes were significantly downregulated. Compared to FFP, there still were 212 genes significantly upregulated and 181 genes significantly downregulated in FFB (Table 1).

Among these upregulated genes, there were some genes common to different treatments. A total of 312 genes were significantly upregulated after *F. fomentarius*-degraded wood, including birch and pine. Compared with *F. fomentarius*-degraded pinewood, 75 significantly upregulated genes were different in *F. fomentarius*-degraded birchwood (Figure 2C). 

Compared with FFC, the clustering heat map clearly shows that FFP and FFB samples were clustered into one group after intra-group aggregation (Figure 2A). There was a significant difference in gene expression between FFC and FFB/FFP, which shows that the addition of wood sticks caused many gene-expression changes. Furthermore, the clustering of FFB and FFP shows the common gene expression in *F. fomentarius*-degraded deadwood. However, there are obviously differently expressed genes between FFB and FFP. The volcano map (Figure 2B) shows that there was an obvious difference between *F. fomentarius*-degraded birchwood and pinewood with the differential expression of genes. The top 35 differentially expressed genes are shown in Figure 2C (*p* < 0.05, log2(fold changer) > 1.2 or <−1.2). The red and blue colors in the clustering heat maps represent the high or low expression of the gene, respectively. Red represents upregulated genes, and blue represents downregulated genes (Figure 2C). 

### 3.4. Functional Annotation of Differentially Expressed Genes

In *F. fomentarius*-degraded wood, differentially expressed genes performed different functions, and their functions were explored with Gene Ontology (GO; http://www.geneontology.org/, accessed on 7 January 2022). The differentially expressed genes (DEGs) were analyzed with GO enrichment analysis, including biological process (BP), cell composition (CC), and molecular function (MF). In the GO enrichment analysis, the DEGs from FFB wood were classified into 9 cellular components, 127 molecular functions, and 46 biological processes. The histogram of the top 20 GO enrichment terms with the lowest *p*-values drawn after the classification of GO items is shown in Figure 3A. These genes are related to cell walls, peroxidase activity, oxidoreductase activity, acting on peroxide as an acceptor, and oxylipin biosynthetic process. In addition, the DEGs of FFP were classified into 6 cellular components, 94 molecular functions, and 54 biological processes. The histogram of the top 20 GO enrichment terms with the lowest *p*-value drawn after the classification of GO items is shown in Figure 3B. These genes are related to oxidoreductase activity, transporter activity, cofactor binding, and integral and intrinsic components of the membrane process. Based on GO enrichment analysis of FFP vs. FFB, differentially expressed genes were mainly related to the significantly enriched MF items of oxidoreductase activity, catalytic activity, galactose oxidase activity, and so on. CC items that were significantly enriched were integral components of the membrane and intrinsic components of the membrane. There were no differentially expressed genes related to enriched BP items between FFB and FFP. Most of the above molecular functions are related to the progress of wood degradation.

### 3.5. Analysis of Different Genes Using KEGG

KEGG pathway enrichment analysis is a method used to analyze gene-expression data, which can help researchers understand the roles and regulatory mechanisms of specific gene sets in organisms. Usually, the main biochemical pathways can be determined through the Kyoto Encyclopedia of Genes and Genomes (KEGG; https://www.kegg.jp/, accessed on 7 January 2022) pathways. The top pathways with the lowest *p*-values (*p* < 0.05) were used to produce a KEGG enrichment bubble diagram, where the ordinate represents the pathway, the abscissa represents the enrichment factor (the number of differences in the pathway is divided by all the numbers), and the circle size indicates the number, where the bluer the color, the smaller the *p*-value. More differentially expressed genes are enriched in the pathways represented by bluer and bigger bubbles. In this study, KEGG enrichment analysis results of differentially expressed genes are shown in the form of a scatter plot. The results of the FFB–FFC comparison show that the main pathways related to differentially expressed genes were multiple types of metabolism, such as starch and sucrose metabolism, glyoxylate and dicarboxylate metabolism, cysteine and methionine metabolism, etc. According to the FFP–FFC comparison, the main pathways related to differentially expressed genes were arginine and proline metabolism, cysteine and methionine metabolism, alanineaspartate and glutamate metabolism, etc. (Figure 3C). Compared with pinewood degradation, the main pathways related to differentially expressed genes in *F. fomentarius*-degraded birchwood were taurine and hypotaurine metabolism, sphingolipid metabolism, tyrosine metabolism, etc. (Figure 3D). There were 11 shared pathways between FFB and FFP. However, the upregulated genes of the same pathway in different treatments were different (Table 2). 

## 4. Discussion

*Fomes fomentarius* is a simultaneous white-rot fungus and one of the main decomposers of coarse wood from different angiosperm tree species, such as *Fagus sylvatica*, *Tilia cordata* or *Betula pendula* etc. [19,20,21]. Gymnosperm tree species, including *Pinus*, *Picea,* and *Abies,* are not its natural hosts. However, *F. fomentarius* can also decompose these gymnosperm tree species’ coarse wood with different degradation characteristics, including deadwood physical and chemical properties, which is due to the different mechanisms and enzyme activities employed by *F. fomentarius* for degrading coarse wood from its natural and unnatural hosts. However, the potential degrading ability of *F. fomentarius* will help it adapt to different host substrates. Maybe this is the main reason why this fungus is widely distributed. 

Weigh loss or mass loss (ML) is a common parameter that can be used to assess the impact of fungal degradation on wood with different wood-decaying fungal species [22,23,24,25]. In this study, under the same culturing conditions, *F. fomentarius*-degraded birch and pinewood sticks separately, and there was an obvious mass loss discrepancy between the experiments. Compared with mass loss in pinewood, the greater mass loss in birch indicates higher degradation efficiency. Cristini et al. [26] also measured the ML caused by *F. formetarius* in birchwood after two months (16.24%); ML loss in birch in our study was 35.42%. In another research work, *Pleurotus ostreatus* and *Trametes versicolor* colonized birchwood for two months, and the ML values were about 27% and 35%, respectively [25]. In contrast, ML in pinewood decayed by *F. fomentarius* was 13.17%, less than that in birchwood.

Cellulose and lignin are the xylem cell-wall components of wood, and their structure is destroyed when wood-decaying fungi attack wood. The wood density distribution, cellulose, and lignin content tend to decrease with longer fungal exposure [26]. Different fungal species colonize different host tree species with different decaying processes [27]. In this study, *F. fomentarius* showed strong degradation abilities, especially when growing in birchwood. Compared with pinewood, in birchwood, although the cellulose degradation rate was lower than that in pine, net cellulose loss is greater due to higher ML loss. 

The ability of wood-decaying fungi to decompose aromatic lignin polymers is possible due to the secretion of organic acids, secondary metabolites, and oxidoreductive metalloenzymes, heme peroxidases, and laccases, encoded by divergent gene families in these fungi [8]. To overcome the lignin barrier in deadwood, white-rot fungi are equipped with varying sets of oxidoreductases (peroxidases and laccase) [28,29]. These oxidative enzymes include extracellular oxidative enzymes (oxidoreductases), i.e., peroxidases and phenol oxidases of the laccase type [8]. Till now, many enzymes have been observed, including laccase, total peroxidase, endocellulase, xylanase etc. Their activities were revealed to be the most important predictors of wood decay [17]. During wood degradation by *F. fomentarius*, lignocellulose-degrading enzymes, including endo-1,4-β-glucanase, 1,4-β-glucosidase, cellobiohydrolase, endo-1,4-β-xylanase, 1,4-β-xylosidase, Mn peroxidase, and laccase were reported to be characterized [30]. After *F. fomentarius* had degraded birch and pinewood for 60 days, seven enzymes were detected to present different enzyme activities in this study. When *F. fomentarius* colonized birchwood sticks, the activities of endo-1,4-β-glucanase, 1,4-β-*N*-acetylglucosaminidase, cellobiohydrolase, and β-1,4-glucosidase were higher than those in birchwood sticks. These enzymes belong to hydrolytic enzymes and cellulosic enzymes [16]. It is worth mentioning that the activity of laccase, as the most common and highly degradable enzyme, in pine deadwood degradation by *F. fomentarius* was much higher than that in birch deadwood degradation. It is conjectured that higher laccase activity in pinewood may be related to the content of pine resin in it because aromatic compounds can be laccase inducers. Maybe this is the main reason that, despite the MS loss in birchwood sticks being higher than that in pinewood sticks, the lignin and cellulose degradation rates in pinewood sticks were high. As its natural host, birch deadwood is degraded more effectively by *F. fomentarius*. The involvement of more peroxidase activity, antioxidant activity, oxidoreductase activity, and hydrolase activity makes degradation more efficient. 

Based on evolutionary analysis, the key genes for lignin degradation ability code for the secretion of Class II peroxidase (POD) enzymes. The assembly and breakdown of carbohydrate polymers and glycoconjugates are carried out by a diverse panel of carbohydrate-active enzymes (CAZymes). CAZymes currently incorporates more than 300 sequence families subdivided into the following classes: glycoside hydrolases (GH), glycosyltransferases (GT), polysaccharide lyases (PL), carbohydrate esterases (CE), and carbohydrate-binding modules (CBM). In addition, oxidoreductases involved in lignin modification and degradation, such as laccases and Class II peroxidases, have been included in the CAZy database as 10 auxiliary activity (AA) families [31]. 

We employed transcriptome analysis to explore these discrepancies in the mechanisms related to wood physical-chemistry characteristics and enzyme activities between FFB and FFP. Compared with hyphae growing in a malt-soaking powder culture medium, there were 1397 differentially expressed genes (DEGs) in FFB and 960 DEGs in FFP. In fact, the same biological process of decomposing deadwood, deriving from either angiosperms or gymnosperms, must be the result of abundant genes that play the same molecular function. In GO enrichment analysis, among the top 50 terms related to these DEGs in FFB and FFP, respectively, there were 31 shared terms, most of them related to wood degradation, such as catalytic activity, oxidoreductase activity, cofactor binding, symporter activity, and gibberellin 2-β-dioxygenase activity. In *F. fomentarius*-led degradation of deadwood from different tree species, the upregulated genes from different families played their different functional roles. As the natural host of *F. fomentarius*, the degradation of FFB involved more DEGs related to terms including peroxidase activity, antioxidant activity, oxidoreductase activity (acting on peroxide as an acceptor and acting on paired donors, with incorporation or reduction of molecular oxygen), and hydrolase activity (hydrolyzing *O*-glycosyl compounds and acting on glycosyl bonds). Of the top GO terms related to upregulated genes, 14 gene families were common to both treatments, including AA2, AA4, AA5, AA7, CBM1, CE4, CE8, GH10, GH16, GH28, GH43, GH5, GH7, and PL4. When *F. fomentarius* colonized pine deadwood, on the contrary, the abovementioned terms were not found, but more metabolic processes and biosynthetic processes were, including cysteine metabolic process, glucan metabolic process, dicarboxylic acid biosynthetic process, glutamate biosynthetic process, etc. However, when considering the molecular function, the related upregulated genes varied in different *F. fomentarius*-degraded deadwood types. For example, for oxidoreductase activity, 156 genes were upregulated in FFB, and only 86 genes were upregulated in FFP. 

In the KEGG enrichment analysis, of the top pathways with a *p*-value < 0.05, there were 11 shared pathways between FFB and FFP, including arginine and proline metabolism, nitrogen metabolism, pentose, and glucuronate interconversions, starch and sucrose metabolism, taurine, and hypotaurine metabolism, etc., which represent the necessary biological progress of *F. fomentarius* to degrade deadwood, regardless of the tree species. During the process of cellulose breakdown, cellobiose is cleaved to glucose by β-1,4-glucosidases. However, according to the KEGG analysis, in *F. fomentarius*-degraded birchwood, the upregulated genes included gene_3019, gene_427, gene_3712, gene_10784, gene_3590, gene_2477, which perform the functions of cleaving cellulose to cellodextrin then to cellobiose or d-glucose and cleaving sucrose to d-glucose, with both functions being related to the starch and sucrose metabolism pathway. On the other hand, in the pathway of starch and sucrose metabolism in *F. fomentarius*-degraded pinewood, the upregulated genes were gene_3845, gene_10784, and gene_8031, which perform the functions of cleaving amylose to starch, cleaving glycogen to a-D-glucose-1P then to cellodextrin to cellobiose, and cleaving β-d-glucoside to d-glucose. Similar results were observed for the pathway of pentose and glucuronate interconversions in FFB and FFP, with different genes involved and different metabolites produced. 

Combining the above information with the variety of wood physical-chemical properties, degrading enzyme characteristics, and gene-expression studies, it may be concluded that *F. fomentarius* can colonize conifer tree wood with relatively low efficiency, as this type of wood is not its natural host. These comparative transcriptome analyses have explored the degradation mechanism of *F. fomentarius* with its potential degrading ability to adapt to more different host substrates. However, more genes belonging to CAZyme families were upregulated in FFB (maybe referring to angiosperm) than in FFP (maybe referring to gymnosperm). However, the key factors causing the difference between the two degradation models have not been found, and the mechanism through which *F. formetarius* selects birch and not pine as its natural host has still not been revealed. This is a speculative reason for wood-decaying fungal host preference. We still have a long way to go in understanding the biochemical reactions as well as fungal metabolism and genetic regulation involved. It was hypothesized that there might be components such as aromatic compounds in pinewood inhibiting the metabolic pathways of some of the genes that were upregulated in FFB, have a degradation function, and were not upregulated in FFP. Additionally, some genes may trigger different pathways to adapt to different substrates. The degradation mechanism of this fungal species needs further research. Some culturing experiments, especially with substance addition, can help to solve these problems.

## Figures and Tables

**Figure 1 jof-10-00196-f001:**
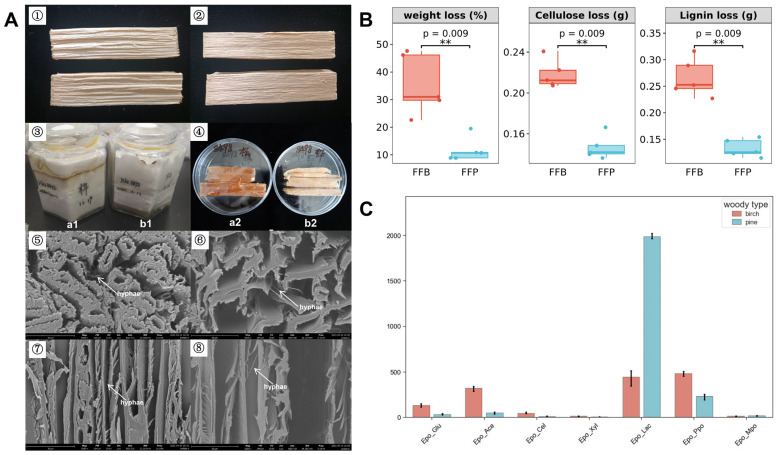
Variations in morphology and physicochemical properties of wood after being degraded by *F. fomentarius* for two months. (**A**). ➀ Pinewood strips before degradation. ➁ Birchwood strips before degradation. ➂ Birch and pinewood sticks culturing in bottles where (**a1**): birch; (**b1**): pine. ➃ Shapes of two kinds of wood sticks after degradation, where. (**a2**): birchwood; (**b2**): pinewood. ➂, ➄ Electron microscope scanning image of a birchwood stick’s cross-section. ➅ Electron microscope scanning image of a pinewood stick’s cross-section. ➆ Electron microscope scanning image of a birchwood stick’s longitudinal section. ➇ Electron microscope scanning image of a pinewood stick’s longitudinal section. Scale bar = 30 μm. (**B**). Weight, cellulose, and lignin loss in wood sticks after degradation by *F. fomentarius* for 2 months. Weight loss expressed as a percentage, and cellulose and lignin loss expressed as quality loss of 1 g wood sample. Asterisks (**) show a significant difference between *F. fomentarius*-degraded birchwood and *F. fomentarius*-degraded pinewood (*p* < 0.01). (**C**). Activities of seven enzymes of *F. fomentarius* after degradation of pine and birchwood sticks for two months. Abbreviation of enzyme names is as follows: Epo_Glu, Endo-1,4-β-glucanase; Epo_Ace, 1,4-β-*N*-acetylglucosaminidase; Epo_Cel, cellobiohydrolase, Epo_Xyl, Endo-1,4-β-xylanase; Epo_Lac, Laccase; Epo_Ppo, β-1,4-glucosidase; Epo_Mpo, β-1,4-xylosidase.

**Figure 2 jof-10-00196-f002:**
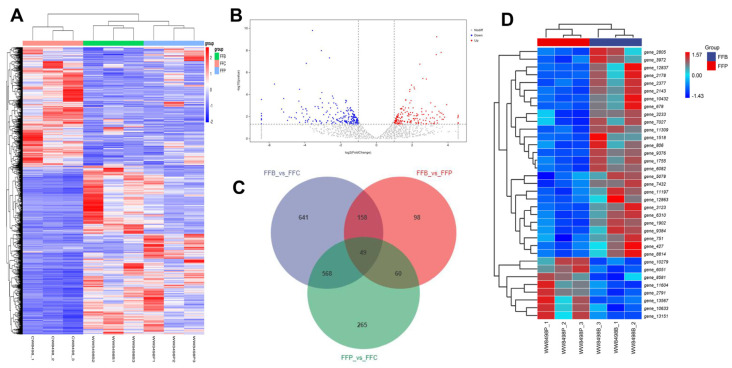
Changes in gene-expression profiles in *F. fomentarius* after different culturing treatments. (**A**) Heat map of differential gene expression between FFC, FFB, and FFP. CW8498−1 refers to (**B**) A volcano map of the number of upregulated and downregulated genes in FFP compared with FFB (*p* < 0.05, log2(fold change) > 1 or <−1). (**C**) A Venn diagram of differentially expressed genes (DEGs) in the transcriptome. The sum of the numbers in each circle represents the total number of DEGs in the comparison combination, and the overlap of the circles represents the common DEGs between the two groups compared. (**D**) Heat map of differential gene expression between FFB and FFP (top 35 genes; *p* < 0.05, log2(fold changer) > 1.2 or <−1.2).

**Figure 3 jof-10-00196-f003:**
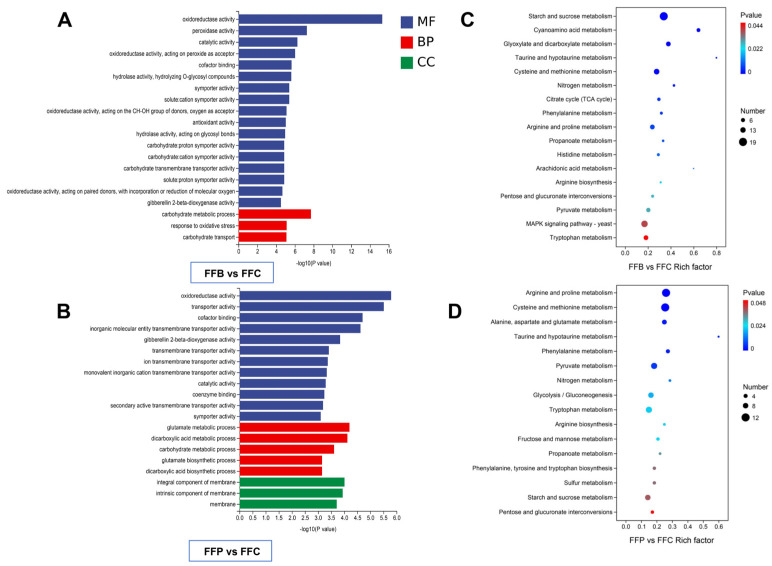
GO and KEGG analyses of the differentially expressed genes in collected hyphae of *F. fomentarius* following different degradation treatments. (**A**,**B**) Quantitative histograms of GO gene enrichment for FFB vs. FFC and FFP vs. FFC. MF: molecular function; BP: biological process; CC: cell composition. The horizontal axis represents GO Term, and the vertical axis represents −log10 (*p*-value) enriched by GO Term. (**C**,**D**) KEGG enrichment bubble charts. The horizontal axis is the rich factor (expressed the ratio of differentially expressed genes number annotated to the pathway/total number of expressed genes annotated to the pathway). The top pathways with the lowest *p*-value were used to produce the map, where the ordinate represents the pathway; the abscissa represents the enrichment factor (the number of differences in the pathway is divided by all the numbers); the circle size indicates the number, where the bluer the color, the smaller the *p*-value. More differentially expressed genes are enriched in the pathways with the bluer and bigger bubbles.

**Table 1 jof-10-00196-t001:** Statistical results of differentially expressed genes.

Treatment	Control	Up Number	Down Number	DEG Number
FFB	FFC	761	636	1397
FFP	FFC	523	437	960
FFB	FFP	212	181	393

**Table 2 jof-10-00196-t002:** Common pathways of up-genes between FFB vs. FFC and FFP vs. FFC.

Common Pathways in FFB and FFP	FFB Upregulated Genes	FFP Upregulated Genes
Arginine and proline metabolism	gene_3060, gene_3216, gene_3728, gene_6398, gene_11214, gene_11215	gene_3060, gene_3728, gene_11214, gene_11215
Arginine biosynthesis	gene_3743	-
Cysteine and methionine metabolism	gene_1945, gene_6207, gene_6296, gene_6974, gene_6361, gene_8194, gene_12255	gene_1628, gene_6974, gene_6977, gene_12255
Nitrogen metabolism	gene_10400, gene_13672	gene_10398, gene_10400
Pentose and glucuronate interconversions	gene_6893, gene_9547, gene_10780, gene_11651	gene_2187, gene_2927, gene_14482
Phenylalanine metabolism	gene_1945, gene_3216, gene_8194, gene_11214	gene_11214
Propanoate metabolism	-	gene_5467
Pyruvate metabolism	gene_1293, gene_6127, gene_8215	gene_5467
Starch and sucrose metabolism	gene_427, gene_2477, gene_3019, gene_3590, gene_3712, gene_10784	gene_3845, gene_8031, gene_10784
Taurine and hypotaurine metabolism	gene_674, gene_2880, gene_6947	gene_674, gene_5467
Tryptophan metabolism	gene_1945, gene_3216, gene_5115, gene_5130, gene_9810, gene_8194, gene_11214	gene_5130, gene_9810, gene_11214

## Data Availability

Data are contained within the article.

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
