# Peer review of "Comparative Transcriptome Analysis Explores the Mechanism of Angiosperm and Gymnosperm Deadwood Degradation by Fomes fomentarius"

_jof, 2024, doi:10.3390/jof10030196_

Round 1
Reviewer 1 Report
The article is devoted to the study of wood decomposition by fungus. Although this fungus has been widely studied, studies of the decay of pine and birch wood may be of scientific interest.
page 2 line 75 - 2.1 F. formentarius cultivation and wood stick degradation It is not indicated how many wood strips were added to one bottle or per liter of the hyphal culture medium?
page 3 line 143 “In the aqueous extracts of the 143 milled samples…” There is insufficient information on the preparation of enzyme extracts. How were the wood samples milled? Was a specific fraction of milled wood selected (particle size)? How much water was used for extraction?
page 4 line 168 “…and the shape of wood sticks changed obviously…” Perhaps one should imagine the appearance of the wood before the experiment.
page 4 lines 171-177 “The dry weight of birch wood …Cellulose content… Lignin content…” In the materials and methods section there is no information about the methods used to carry out the measurements. This information should be added.
page 4 line 180 “activity of …. 1,4-β-N-acetylglucosaminidase….” Why was the activity of this enzyme measured? It is associated with the metabolism of chitin, not wood.
page 4 lines 182 – 183 Enzyme activity is indicated as nmol/g/min. In the section “2.4 Degradative Enzyme Activities”, the one unit of enzyme activity is defened as µM/min. In Figure 5c the activity is also indicated as nmol/g/min. Activity should be described as U/g. Why the activity of laccase was measured? F. formentarius produces another ligninolytic enzyme - manganese peroxidase.
page 10 lines 351-353 Could the higher laccase activity in pine wood be related to the content of pine resin in it? Aromatic compounds can be laccase inducers.
Author Response
Reviewer 1:
page 2 line 75 - 2.1 F. formentarius cultivation and wood stick degradation It is not indicated how many wood strips were added to one bottle or per liter of the hyphal culture medium?
These details has been added in the text.
page 3 line 143 “In the aqueous extracts of the 143 milled samples…” There is insufficient information on the preparation of enzyme extracts. How were the wood samples milled? Was a specific fraction of milled wood selected (particle size)? How much water was used for extraction?
These details has been added in the text.
page 4 line 168 “…and the shape of wood sticks changed obviously…” Perhaps one should imagine the appearance of the wood before the experiment.
The photo of origin wood stripe has been added in figure 1.
page 4 lines 171-177 “The dry weight of birch wood …Cellulose content… Lignin content…” In the materials and methods section there is no information about the methods used to carry out the measurements. This information should be added.
These details has been added in the text.
page 4 line 180 “activity of …. 1,4-β-N-acetylglucosaminidase….” Why was the activity of this enzyme measured? It is associated with the metabolism of chitin, not wood.
The selection of enzyme came from reference 15 which reported the related wood-degrading enzyme types.
page 4 lines 182 – 183 Enzyme activity is indicated as nmol/g/min. In the section “2.4 Degradative Enzyme Activities”, the one unit of enzyme activity is defened as µM/min. In Figure 5c the activity is also indicated as nmol/g/min. Activity should be described as U/g. Why the activity of laccase was measured? F. formentarius produces another ligninolytic enzyme - manganese peroxidase.
The indication of enzyme activity came from reference 15.
page 10 lines 351-353 Could the higher laccase activity in pine wood be related to the content of pine resin in it? Aromatic compounds can be laccase inducers.
Thank for this suggestion and has been added in the text.
Reviewer 2 Report
In their manuscript, Wei et al. describe the comparative analysis of the degradation of wood from different trees by the white rot basidiomycete Fomes fomentarius. They performed biochemical, enzymatic and transcriptome analyses on birch and pine wood degraded by the fungus. The results are of interest to researchers interested in the bioconversion of plant materials and fungal metabolism. However, there are a few points where the manuscript should be improved which are given in the detailed comments section.
Points where the manuscript should be improved:
1. Line 79: Please give more information on the wood that was used for these experiments (e.g. were the pieces of wood derived from plant structures of the same age, were they taken from the outer layers or from the core of the wood etc.).
2. Line 118: Please give the accession number and literature reference of the genome that was used for mapping.
3. Line 161: The BioProject that is mentioned contains 9 accession numbers (SRX22152964-SRX22152972), but there is no information what sample (i.e. what type of wood or other substrate) was analyzed in which sample. Please add a table to the manuscript describing for each accession what was sequenced. Please also add the information if these samples include independent biological replicates.
4. Figure 1A: It would be better to have images of the wood not only after but also before degradation. Also, please increase the size of the labelling of the scale bar. Also, please indicate examples in the Figure of where the fungal mycelium that is mentioned in the text can be seen.
5. Lines 171-177 and Figure 1B: While the weight loss is much greater in degraded birch wood than pine wood, this is not the case when looking at cellulose and lignin loss. Please explain what mechanism would explain this difference. It may also be easier to understand if you give actual measurements (e.g. in gram, not percentages).
6. Line 223: Should it be 312 (not 302)? This is what you end up with when adding up the numbers in Figure 2B.
7. Lines 228-234: None of the obervations described in the text can be seen in Figures 2A or 2C. Please describe more clearly what was done and/or add a different Figure.
8. Line 250 and Figure 2B: In the text, it says "Venn diagram of differentially expressed genes", but the numbers suggest that these are only the upregulated genes.
9. Figure 3A: Please explain the meaning of the abbreviations MF, BP and CC in the Figure.
10. Figure 3C and D and Lines 281-282: It is not clear what the x-axis means. Enrichment factor would imply values larger than 1, but all values in the diagram are <1. In the Figure legend in lines 281-282 it is not clear what "numbers" refers to. Please clarify.
11. The species name is misspelled in several instances in the manuscript ("formentarius" instead of "fomentarius").
Author Response
Points where the manuscript should be improved:
Line 79: Please give more information on the wood that was used for these experiments (e.g. were the pieces of wood derived from plant structures of the same age, were they taken from the outer layers or from the core of the wood etc.).
These details has been added in the text.
Line 118: Please give the accession number and literature reference of the genome that was used for mapping.
The literature reference genome website has been added in the text.
Line 161: The BioProject that is mentioned contains 9 accession numbers (SRX22152964-SRX22152972), but there is no information what sample (i.e. what type of wood or other substrate) was analyzed in which sample. Please add a table to the manuscript describing for each accession what was sequenced. Please also add the information if these samples include independent biological replicates.
The information of RNA sample has added in 2.1 F. fomentarius cultivation and woody stripe degradation
Figure 1A: It would be better to have images of the wood not only after but also before degradation. Also, please increase the size of the labelling of the scale bar. Also, please indicate examples in the Figure of where the fungal mycelium that is mentioned in the text can be seen.
The details has been added in figure 1.
Lines 171-177 and Figure 1B: While the weight loss is much greater in degraded birch wood than pine wood, this is not the case when looking at cellulose and lignin loss. Please explain what mechanism would explain this difference. It may also be easier to understand if you give actual measurements (e.g. in gram, not percentages).
Thank for the suggestion and the cellulose and lignin loss have been expressed with gram.
Line 223: Should it be 312 (not 302)? This is what you end up with when adding up the numbers in Figure 2B.
Yes, it is 312.
Lines 228-234: None of the obervations described in the text can be seen in Figures 2A or 2C. Please describe more clearly what was done and/or add a different Figure.
The obervations described has been added.
Line 250 and Figure 2B: In the text, it says "Venn diagram of differentially expressed genes", but the numbers suggest that these are only the upregulated genes.
We modified the Venn diagram with differentially expressed genes.
Figure 3A: Please explain the meaning of the abbreviations MF, BP and CC in the Figure.
We explained in legend of Figure 3A.
Figure 3C and D and Lines 281-282: It is not clear what the x-axis means. Enrichment factor would imply values larger than 1, but all values in the diagram are <1. In the Figure legend in lines 281-282 it is not clear what "numbers" refers to. Please clarify.
The means of x-axis has given and explain the enrichment factor is the ratio of differentially expressed genes number annotated to the pathway/total number of expressed genes annotated to the pathway. So it is <1.
The species name is misspelled in several instances in the manuscript ("formentarius" instead of "fomentarius").
Thanks very much and it has all modified.
Reviewer 3 Report
The article submitted to me for review was entitled: Comparative transcriptome analysis investigates the mechanism of degradation of dead angiosperms and gymnosperms by Fomes fomentarius. it is undoubtedly an interesting paper. However, I feel that it could have been written in more detail to make it easier to understand for future readers. Below I have added some comments and notes on individual parts of the text. By addressing these points, I believe you can strengthen your manuscript by providing a comprehensive interpretation of the results and their broader implications in the field of mycology and forest ecology.
Summary:
- Explicitly state the aim or hypothesis of the study.
- Mention the significance or possible implications of the results.
Introduction:
- Explain the importance of understanding the decomposition mechanism of F. formentarius, especially in the context of forest ecosystems and the carbon cycle.
- Explain the novelty or knowledge gap that this study aims to fill.
- Consider citing current literature to support the importance of transcriptome profiling in understanding fungal mechanisms.
Materials and methods:
2.1 F. formentarius cultivation and decomposition of woodblocks:
- Provide details of the environmental conditions during cultivation (e.g. humidity).
- Indicate how many replicates were performed for each condition and whether these were randomised.
2.3 Baseline analysis and mRNA analysis of the F. formentarius transcriptome:
- Explain the rationale for the choice of parameters for differential expression analysis.
- Indicate the number of replicates used for differential expression analysis.
2.4 Degradable enzyme activities:
- Explain the method used to extract the enzymes from the samples.
2.5 Statistical analysis:
- Explain the rationale for the choice of statistical tests used.
- Indicate whether the statistical analyses were performed on raw or transformed data.
Results:
3.1 Decomposition of wooden sticks:
- Give details of the experimental design, including the number of replicates and any controls used.
- Explain the method you used to assess the destruction of the wood structure and the growth of hyphae inside the wood.
3.3 Analyse the differential expression of mRNA:
- Consider a brief explanation of volcano curves and heat maps for readers unfamiliar with these visualisation techniques.
3.5 Analyse different genes using KEGG:
- Justify the selection of KEGG pathways for analysis and explain their relevance to wood decomposition in F. formentarius.
- Explain whether the identified pathways are consistent with previous studies or with expectations based on fungal metabolism.
Discussion:
- Provide a brief overview of Fomes formentarius, including its ecological role and importance as a wood-degrading fungus.
- Explain the importance of studying wood decomposition by F. formentarius in the context of forest ecology, the carbon cycle and fungal biology.
- Justify the choice of mass loss as a key parameter for the assessment of wood decomposition and discuss its importance in ecological and industrial contexts.
- discuss the importance of transcriptome analysis in elucidating the molecular mechanisms underlying wood decomposition by F. formentarius.
- provide a more detailed discussion of the significance of the identified GO terms and KEGG pathways in the context of wood degradation by fungi.
- Investigate hypotheses and potential mechanisms underlying the observed host preference of F. formentarius for birch over pine.
- Discuss the implications of understanding fungal host preference for forest ecology, fungal biotechnology, and sustainable forest management practises.
- Propose future research directions, including experimental approaches and analytical techniques, to further elucidate the degradation mechanisms and metabolic pathways involved.
Author Response
Summary:
- Explicitly state the aim or hypothesis of the study.
These details has been added in the text.
- Mention the significance or possible implications of the results.
Introduction:
- Explain the importance of understanding the decomposition mechanism of F. formentarius, especially in the context of forest ecosystems and the carbon cycle.
The importance has been added in the text.
- Explain the novelty or knowledge gap that this study aims to fill.
We added the study aims with question and the answer.
- Consider citing current literature to support the importance of transcriptome profiling in understanding fungal mechanisms.
The current literature has been cited (reference 13-2023, reference 14-2022).
Materials and methods:
2.1 F. formentarius cultivation and decomposition of woodblocks:
- Provide details of the environmental conditions during cultivation (e.g. humidity).
These details has been added in the text.
- Indicate how many replicates were performed for each condition and whether these were randomised.
These details has been added in the text.
2.3 Baseline analysis and mRNA analysis of the F. formentarius transcriptome:
- Explain the rationale for the choice of parameters for differential expression analysis.
These details has been added in the text.
- Indicate the number of replicates used for differential expression analysis.
These details has been added in the text.
2.4 Degradable enzyme activities:
- Explain the method used to extract the enzymes from the samples.
These details has been added in the text.
2.5 Statistical analysis:
- Explain the rationale for the choice of statistical tests used.
These details has been added in the text.
- Indicate whether the statistical analyses were performed on raw or transformed data.
These details has been added in the text.
Results:
3.1 Decomposition of wooden sticks:
- Give details of the experimental design, including the number of replicates and any controls used.
These details has been added in the text.
- Explain the method you used to assess the destruction of the wood structure and the growth of hyphae inside the wood.
These details has been added in the text.
3.3 Analyse the differential expression of mRNA:
- Consider a brief explanation of volcano curves and heat maps for readers unfamiliar with these visualisation techniques.
A brief explanation has added in the text.
3.5 Analyse different genes using KEGG:
- Justify the selection of KEGG pathways for analysis and explain their relevance to wood decomposition in F. formentarius.
A brief description has added in the text.
- Explain whether the identified pathways are consistent with previous studies or with expectations based on fungal metabolism.
Sorry for I couldn’t find the previous literature with these pathways at present. I will keep in looking for.
Discussion:
- Provide a brief overview of Fomes formentarius, including its ecological role and importance as a wood-degrading fungus.
A brief overview has added in the text.
- Explain the importance of studying wood decomposition by F. formentarius in the context of forest ecology, the carbon cycle and fungal biology.
A brief description has added in the text.
- Justify the choice of mass loss as a key parameter for the assessment of wood decomposition and discuss its importance in ecological and industrial contexts.
A brief description has added in the text.
- discuss the importance of transcriptome analysis in elucidating the molecular mechanisms underlying wood decomposition by F. formentarius.
Yes, we added the discussion.
- provide a more detailed discussion of the significance of the identified GO terms and KEGG pathways in the context of wood degradation by fungi.
Yes, we added the discussion.
- Investigate hypotheses and potential mechanisms underlying the observed host preference of F. formentarius for birch over pine.
We added the hypotheses.
- Discuss the implications of understanding fungal host preference for forest ecology, fungal biotechnology, and sustainable forest management practises.
Yes, we discussed.
- Propose future research directions, including experimental approaches and analytical techniques, to further elucidate the degradation mechanisms and metabolic pathways involved.
We added some suggestion about experiment.
Round 2
Reviewer 3 Report
The authors have followed all my comments, which has considerably improved the clarity of the article. In its present form, the article is suitable for publication in JoF.
non